# Spatial intimacy of binary active-sites for selective sequential hydrogenation-condensation of nitriles into secondary imines

Sai Zhang[1,3], Zhaoming Xia[2,3], Yong Zou[2], Mingkai Zhang[2] & Yongquan Qu [1,2✉]

Precisely controlling the spatial intimacy of multiple active sites at sub-nanoscale in heterogeneous catalysts can improve their selectivity and activity. Herein, we realize a highly selective nitrile-to-secondary imine transformation through a cascaded hydrogenation and condensation process by $Pt_1/CoBO_x$ comprising the binary active sites of the single-dispersed Pt and interfacial Lewis acidic B. Atomic Pt sites with large inter-distances (>nanometers) only activate hydrogen for nitrile hydrogenation, but inhibit condensation. Both adjacent B...B on $CoBO_x$ and neighbouring Pt...B pairs with close intimacy of ~0.45 nm can satisfy the spatial prerequisites for condensation. Mechanism investigations demonstrate the energetically favorable pathway occurred on adjacent Lewis acidic B sites through the nitrile adsorption (acid-base interaction), hydrogenation via hydrogen spillover from Pt to B sites and sequential condensation. Strong intermolecular tension and steric hindrance of secondary imines on active sites lead to their effective desorption and thereby a high chemoselectivity of secondary imines.

[1] Key Laboratory of Special Functional and Smart Polymer Materials of Ministry of Industry and Information Technology, School of Chemistry and Chemical Engineering, Northwestern Polytechnical University, Xi'an, China. [2] Center for Applied Chemical Research, Frontier Institute of Science and Technology, Xi'an Jiaotong University, Xi'an, China. [3]These authors contributed equally: Sai Zhang, Zhaoming Xia ✉email: yongquan@nwpu.edu.cn

The ideal scenario for heterogeneous catalysis is to solely proceed to a single product with high activity and stability[1]. However, rare cases can realize this ultimate target, especially for catalytic selectivity. Generally, several pathways or elementary steps controlled by thermodynamics and kinetics could sequentially and/or parallelly take place on nanoscaled catalyst surfaces owing to the possible interaction of metal sites with each functional group (Fig. 1a), therefore resulting in a poor selectivity. Recently, reducing the size of catalytically active components has been demonstrated to effectively improve the catalytic selectivity by controlling the adsorption events on active sites of catalysts (Fig. 1b). Especially, the metal single-atom catalysts (SACs) have exhibited the improved catalytic activity and selectivity for various important reactions, including $CO_2$ transformation, water–gas shift reaction, semi-hydrogenation of acetylene, and hydrogenation of nitroarenes[2–4].

However, many catalytic reactions generally require multiple cascaded steps with several reactant molecules and/or intermediates on their respective active sites. As a typical cascaded reaction, the selectively sequential hydrogenation–condensation of nitriles into secondary imines is highly desirable since it represents a green and cost-effective one-step process to replace the previously traditional methods[5–7]. Generally, four possible distinct processes are involved (Fig. 1c): (1) hydrogenation of nitriles into primary imines, (2) and then into primary amines, (3) nucleophilic attack on the electron-deficient carbon of primary imines by primary amines to produce secondary imines, and (4) undesired over-hydrogenation of secondary imines[8,9]. Previous reports have proved that all these elementary steps take place on catalysts surface[10]. Thus, hydrogenation of several nitrile molecules can simultaneously occur on the surface of a metal nanoparticle, which is beneficial for the condensation process to give secondary imines. However, severe over-hydrogenation of secondary imines to secondary amines is usually inevitable due to the strong adsorption of secondary imines on metal nanoparticles[11–15]. While, the primary amines were also obtained in the presence of acids or alkaline additives, which could suppress the occurrence of Step 3[16–19]. By examining plenty of literature, the chemoselectivity transformation of nitrile into secondary imines is still lacked fundamental understandings and faces great scientific challenges. Thus, to design a heterogeneous catalyst with high activity and selectivity for the selective nitrile-to-imine transformation is highly needed.

Theoretically, the metal SACs might avoid the over-hydrogenation of secondary imines due to the weak interaction between reactants and small size metals, especially those of sub-nanometer clusters and SACs[20,21]. However, the isolated metal single-atom active sites with the large inter-distances are only occupied by one reactant molecule (Fig. 1b), thereby making the condensation step difficult and inhibiting the occurrence of Step 3. Taking the above analysis into considerations, if another active site is introduced with a well-organized spatial distribution, the metal SACs might break this limitation to achieve the selective nitrile-to-imine transformation. The Lewis basic $N$ atom in nitriles and the hydrogenated intermediates have inspired the use of Lewis acidic sites to trap reactants and intermediates through a strong acid–base interaction.

In this work, the atomically dispersed Pt in $CoBO_x$ nanosheets ($Pt_1/CoBO_x$) has been designed to successfully realize the highly selective nitrile-to-imine transformation with high activity and stability in the absence of any additives. The $CoBO_x$ nanosheets were selected owing to their abundance of the interfacial Lewis acidic B sites to trap nitriles as well as hydrogenated intermediates. In the $Pt_1/CoBO_x$ catalyst, three spatial configurations

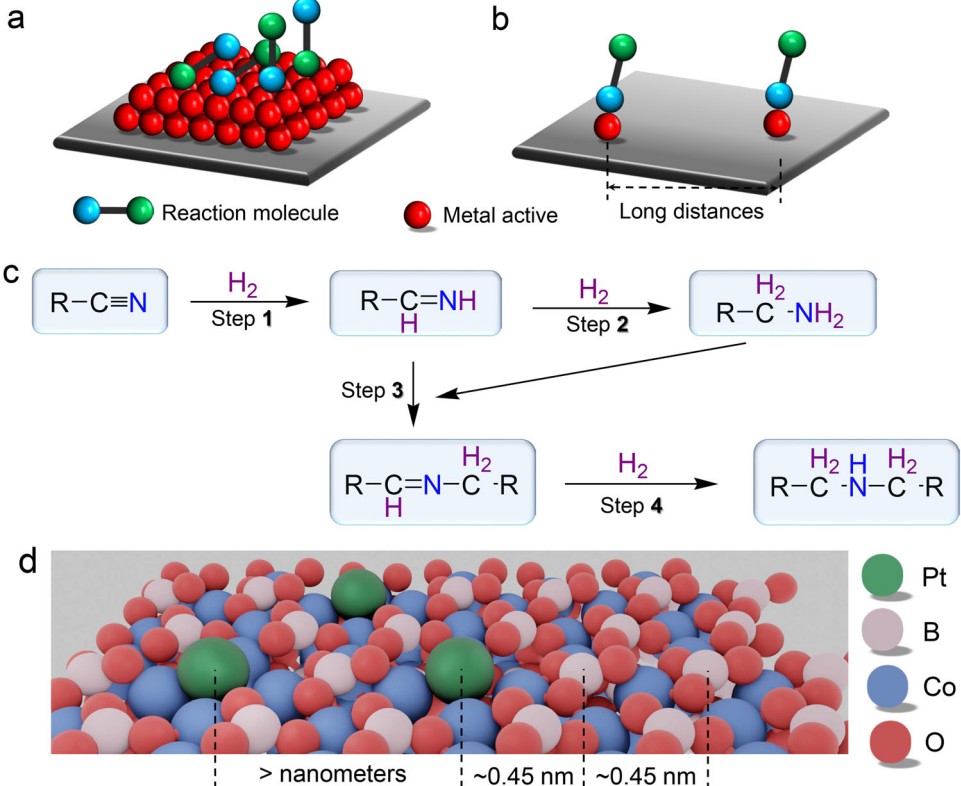

**Fig. 1 Illustration of the spaital distribution of reactants on various catalytic active sites.** Spatial configuration of reactants with binary functional groups on **a** metal nanoparticles, and **b** metal single-atom catalyst sites. **c** Possible pathways of nitrile hydrogenation. **d** Spatial intimacy of binary active-sites of single-atom Pt sites and interfacial Lewis acidic B.

of binary active sites of single-dispersed Pt site and interfacial Lewis acidic B site are illustrated in Fig. 1d: Pt…Pt with a large inter-distance over nanometers; neighboring Pt…B and adjacent B…B site with a distance of ~0.45 nm. Since the strong adsorption of nitriles and intermediates on single-dispersed Pt sites, theoretically, only the latter two couples can allow the successful condensation if considering their spatial configurations. Catalytic mechanism investigations show that the atomically dispersed Pt sites guarantee efficient hydrogen dissociation. Afterward, the efficient hydrogenation of the absorbed nitriles on adjacent Lewis acidic B sites through hydrogen spillover from Pt to B and thereby subsequent condensation (Step 3) are the energetically favorable pathways. Such a catalyst with the spatially organized single-dispersed Pt and Lewis acidic B binary active-sites can yield a >99.9% selectivity and $864\,h^{-1}$ TOF of benzonitrile hydrogenation to corresponding secondary imines.

## Results

**Preparation and characterization of the $Pt_1/CoBO_x$ catalyst.** The $Pt_1/CoBO_x$ catalyst with 0.79 wt% Pt-loading was prepared by a facile wet chemical process. Darkfield transmission electron microscopy (TEM) image showed an ultrathin nanosheet morphology of catalyst without any apparent metallic particles/clusters (Fig. 2a). Meanwhile, energy dispersive spectroscopy revealed the uniform Pt distribution in the nanosheets (Supplementary Fig. 1), indicating the highly dispersed Pt on $CoBO_x$. Also, the high-angle annular dark-field scanning TEM (HAADF-STEM) image further revealed the atomically isolated Pt species were throughout the whole nanosheets (Fig. 2b).

However, different from the oxidized Pt in previously reported SACs[22–26], X-ray photoelectron spectrum (XPS) suggested a near metallic state of Pt in $Pt_1/CoBO_x$ catalyst (Supplementary Fig. 2). This phenomenon could be attributed to the reductive environment of synthetic solution in the presence of $NaBH_4$ as well as the much easier reducibility of $Pt^{2+}$ ions than that of $Co^{2+}$ ions according to their standard reduction potentials. To clearly elucidate the Pt chemical status, X-ray absorption fine spectroscopy (XAFS) was applied. X-ray absorption near-edge structures of Pt K-edge revealed that the white line peak of the $Pt_1/CoBO_x$ catalyst located at 11568.4 eV was between those of $Pt^0$ (11568.0 eV) and $PtO_2$ (11569.8 eV), but very close to that of $Pt^0$ (Fig. 2c). Also, the peak area of the white line peak was slightly larger than that of $Pt^0$. These results illustrated that the electronic structure of Pt was close to the metallic state, consistent with XPS results.

Extended XAFS (EXAFS) spectra were acquired to provide the chemical environment of Pt. Compared with $PtO_2$, the peak of catalyst at ~1.6 Å was labeled as Pt–O bond (Figs. 2d, e). The second peak of catalyst between 2.2 and 3.2 Å was well fitted with Pt–Co scattering path (Table 1 and Supplementary Fig. 3a), rather than Pt–Pt or Pt–B scattering path with large R-factors (Supplementary Table 1 and Supplementary Fig. 3b, c). Thus, the negligible Pt–Pt coordination suggested bare Pt particles/clusters in catalysts. Wavelet transform of EXAFS also showed that EXAFS of this catalyst was mainly contributed by Pt–O, and Pt–Co pathways (Supplementary Fig. 4). The averaged coordination number (CN) of Pt–Co was ~4.7, indicating that one Pt atom was chemically bonded with 4–5 Co atoms with an average bond length of 2.61 Å.

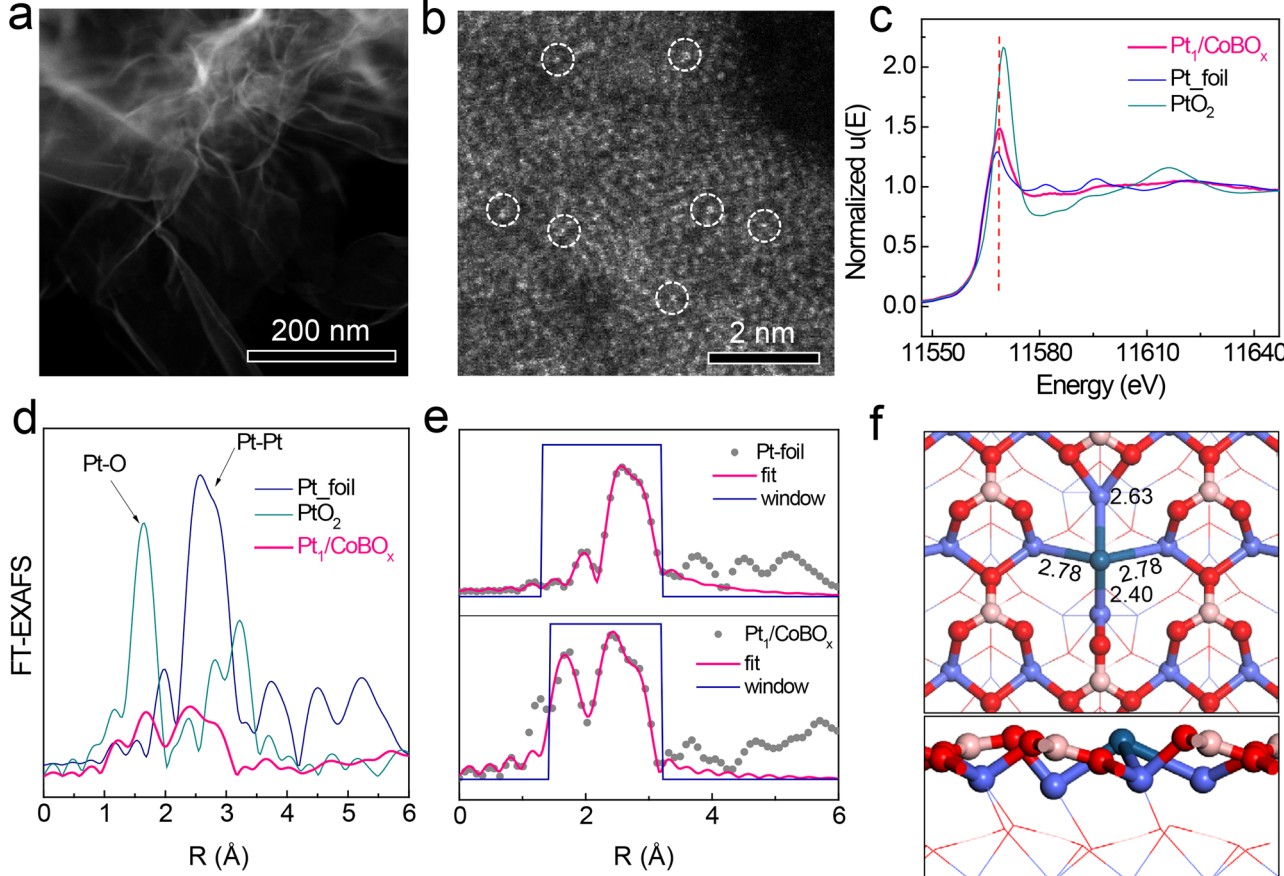

**Fig. 2 Characterizations of the $Pt_1/CoBO_x$ catalyst. a** Darkfield TEM and **b** HAADF-STEM images of $Pt_1/CoBO_x$. **c** XANES and **d** EXAFS of $Pt_1/CoBO_x$, Pt foil, and $PtO_2$ powder. ($k^3$ weighted data). **e** Curve fittings of EXAFS data. **f** Calculated model of $Pt_1/CoBO_x$. The cyan, blue, red, and pink balls represent the Pt, Co, O, and B atoms, respectively.

**Table 1 Structural parameters of catalysts and Pt foil from the EXAFS fittings.**

|          |       | CN (a.u.)      | R (Å)         | $\sigma^2$ ($10^{-3}$) | R-factor |
|----------|-------|----------------|---------------|------------------------|----------|
| Pt-foil  | Pt-Pt | 12[a]          | 2.76 ± 0.01   | 3.9 ± 0.1              | 0.002    |
| $Pt_1/CoBO_x$ | Pt-O  | 1.6 ± 0.8 | 2.04 ± 0.05   | 4.0 ± 1.1              | 0.004    |
|          | Pt-Pt | 0.2 ± 0.6      | 2.72 ± 0.16   | 3.0 ± 2.7              |          |
|          | Pt-Co | 4.7 ± 3.0      | 2.61 ± 0.03   | 7.3 ± 4.9              |          |

[a]amp = 0.826 ± 0.04.
CN is the averaged coordination number of each shell.
R is half of the averaged distance of the scattering path.
$\sigma^2$ is the mean square variation in path length.

Meanwhile, due to the low weight percentage of Pt and the interference of other elements, DFT simulation was used to further confirm the local structure of Pt in the $Pt_1/CoBO_x$ catalysts, which could exclude the possibility of wrong structure caused by fitting error. Theoretical models of catalysts were constructed by a Pt atom and $Co_3(BO_3)_2(200)$ slab to examine the XFAS-derived structures. The single Pt atom can be adsorbed on the surface or doped by replacing surface Co, O, B atoms, or one $BO_3$ unit (Fig. 2f and Supplementary Fig. 5). Among them, the one Pt-replaced-$BO_3$ is the most stable structure with the most negative formation energy (Supplementary Fig. 5). The simulated bond lengths of Pt–Co (2.40–2.78 Å, the average value of 2.65 Å) were highly consistent with the EXAFS fittings (Pt–Co: 2.61 Å). Also, one Pt atom was bonded with four Co atoms herein (Fig. 2f), leaving the top of Pt for binding with oxygen, which was consistent with the fitted CN of Pt–Co (4.7) and Pt–O (1.6, Table 1). Thus, the single-atom Pt bonded with 4–5 Co atoms on $CoBO_x$ could be verified by combining experimental and DFT data.

The structural features of $Pt_1/CoBO_x$, revealed from both the experimental and theoretical results, strongly suggest the formation of the unique local configuration of single-atom Pt with four Co atom ($Pt_1$–$Co_4$ sites) in the catalyst. However, those Co atoms with the saturated coordination states are located at the subsurface or the second atom layer (Fig. 2f). Thus, Co atoms in the clusters are unsuitable to serve as the active sites due to the large steric hindrance and the saturated coordination environment, thereby denying the accessibility of nitriles onto Co sites and leaving only Pt sites interacting with nitriles. However, the condensation (Step 3 in Fig. 1c) is also difficult to occur on two atomically dispersed Pt sites owing to their large distance. Luckily, the singly dispersed Pt site is surrounded by several Lewis acidic B sites, giving a distance of 0.45 nm (Fig. 1d and Fig. 2f). Besides, all interfacial Lewis acidic B sites are adjacent with a short distance of 0.45 nm. Generally, Lewis acid is the catalyst for the condensation step[27,28]. Therefore, combining the atomically dispersed Pt sites for $H_2$ activation and the interfacial Lewis acidic sites for condensation, the dual active sites in $Pt_1/CoBO_x$ with three spatial configurations might break the limitation of adsorption in the metal SACs and catalytically transform nitriles into secondary imines.

**Catalytic performance of $Pt_1/CoBO_x$ catalyst.** Inspired by such well-organized spatial distribution of binary active sites in $Pt_1/CoBO_x$, hydrogenation of benzonitrile was selected as a model reaction and optimized at 90 °C and 1 MPa $H_2$ in isopropanol. As a reference catalyst, Pt nanoparticles on carbon black (Pt/C, 0.8 wt%, 4.6 ± 0.7 nm) were prepared by coprecipitation method (Supplementary Fig. 6). The obtained Pt/C catalyst yielded a 97.6% conversion of benzonitrile after 22 h (Fig. 3a) with a TOF value of 569 h$^{-1}$ based on each exposed Pt atom. However, serious over-hydrogenation was observed (Fig. 3b). The final product was dibenzylamine (95%) with very low yields of secondary imines N-benzylidenebenzylamine (1.1%) and benzylamines (2.6%) (Fig. 3c).

Comparatively, the $Pt_1/CoBO_x$ catalysts delivered much higher activity under identical conditions, reaching 98.4% conversion of benzonitrile after 10 h (Fig. 3a). Meanwhile, the TOF value based on each Pt atom (864 h$^{-1}$) was 1.5 times higher than that of Pt/C (569 h$^{-1}$). Most importantly, the selectivity of N-benzylidene-benzylamine was significantly improved to 99.9% (Fig. 3b). While, the yields of benzylamine and N-benzylideneamine were below the detection limitations of gas chromatography–mass spectrometry (GC–MS) (Fig. 3c). In addition, the $Pt_1/CoBO_x$ catalysts also exhibited the preserved performance at least for four consecutive cycles (Fig. 3d). After the reaction, the $Pt_1/CoBO_x$ catalyst could be easily recycled by centrifugal separation and reused for the next cycle without any treatment. Also, the concentration of Pt in the reaction solution was 30 ppm by inductively coupled plasma optical emission spectrometer analysis. Thus, only 0.5 wt% of Pt (relative to the initial Pt amount in $Pt_1/CoBO_x$) was leached from the $Pt_1/CoBO_x$ catalyst after four cycles of the repeatedly catalytic reactions, revealing a very low metal loss and highly structural robustness of catalysts during cycling. Meanwhile, the spent catalysts reserved the initial morphological features. Therefore, the structural robustness of $Pt_1/CoBO_x$ could indicate their catalytic stability during the hydrogenation of benzonitrile (Supplementary Fig. 7).

**Mechanism investigations.** Compared with the Pt/C catalyst, $Pt_1/CoBO_x$ exhibited the greatly improved catalytic selectivity of imines as well as activity. Previous reports have proved that alloy nanoparticles can drive the product distribution shift, especially for the introduction of Co for hydrogenation of nitriles[29–31]. However, the unique geometrical configuration of $Pt_1$–$Co_4$ sites in $Pt_1/CoBO_x$ suggests that the introduction of Co does not play a critical function in the selective transformation of benzonitriles to secondary imines. The detailed reasons are listed as follows: (1) the huge steric hindrance on $Pt_1$–$Co_4$ sites makes it difficult to bind with two reactant molecules simultaneously; (2) Co atoms in $Pt_1$–$Co_4$ sites are located in the subsurface rather than the catalyst surface; (3) Co atoms are saturated by O and Pt atoms, leading to the blocked sites to bind with other reactant molecules. However, the integration of single-atom Pt and four Co atoms can improve the capability for hydrogen activation[32,33].

Objectively, it is difficult to precisely synthesize the $Pt_1$–$Co_4$ clusters on other supports and experimentally demonstrate their influence on the catalytic performance. Herein, the PtCo/C catalysts with 1:4 atomic ratio of Pt:Co and 1 wt% Pt-loading (Supplementary Fig. 8a) were chosen as the nearest approximation model catalyst to understanding the roles of Co incorporated with Pt on the catalytic behavior. As shown in Supplementary Fig. 8b, the measured lattice spacing of 0.215 nm from the HRTEM image of PtCo/C was smaller than the Pt(111) crystalline plane (0.227 nm), but larger than the Co (111) plane (0.205 nm), which indicated the successful formation of the alloyed PtCo nanoparticles on the surface of the carbon. As shown in Fig. 3, the PtCo/C catalyst exhibited the improved catalytic activity compared with the Pt/C catalyst, which was still much slower than that of the $Pt_1/CoBO_x$ catalyst. Most importantly, the selectivity of PtCo/C towards secondary imines N-benzylidenebenzylamine was similar to Pt/C and greatly lower than that catalyzed by $Pt_1/CoBO_x$ at the end of the reaction under the same conditions. While the selectivity of benzylamines was increased to 30% for the PtCo/C catalysts. Obviously, integration of Co and Pt did not shift the selectivity of secondary imines herein. Therefore, this experimental evidence proved that the $Pt_1$–$Co_4$ sites in $Pt_1/CoBO_x$ were not the main factor for the selective transformation of benzonitriles into secondary imines.

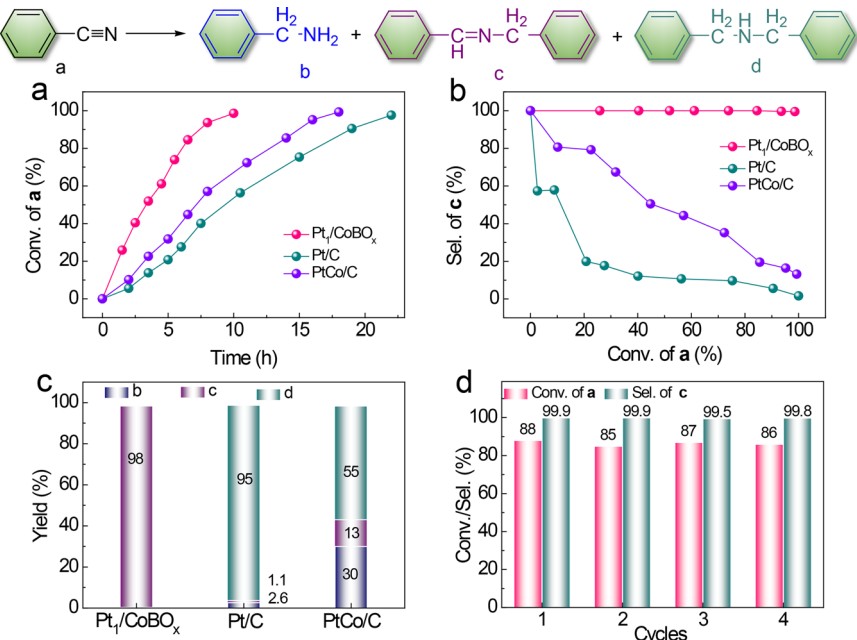

**Fig. 3 Catalytic performance of the Pt$_1$/CoBO$_x$, Pt/C, and PtCo/C catalysts. a** Time course of benzonitrile conversions. **b** Benzonitrile conversion *vs.* selectivity of secondary imines. **c** Final yield of various products. **d** Stability of Pt$_1$/CoBO$_x$. Reaction conditions: benzonitrile (1 mmol), isopropanol (2 mL), catalysts (5 mg), 90 °C, and 1 MPa H$_2$. The reaction time of the stability test was 8 h.

Actually, the catalytic performance of Pt/C and PtCo/C catalyst experimentally revealed that transformation of nitriles into secondary amines on Pt or PtCo nanoparticles suffered from the severe over-hydrogenation into secondary amines, similar to previous reports[34,35]. It's generally attributed to the strong adsorption of imines on the large metal particles[36]. Thus, the hydrogenation pathway of Pt$_1$/CoBO$_x$ with the simultaneously enhanced catalytic activity and selectivity is different from the particle counterparts. To examine the importance of the spatial distribution of binary active sites as well as their proximity, density functional theory (DFT) calculations were performed to examine the adsorption behavior of various molecules on Pt$_1$/CoBO$_x$. Benzonitrile is adsorbed on the Pt top with an adsorption energy of −2.92 eV via N atom interaction with Pt (Fig. 4a and Supplementary Fig. 9). The respective adsorption energies of N-benzylideneamine and benzylamine on Pt$_1$ site are −3.23 and −3.18 eV with similar adsorption configurations. Such strong adsorption of intermediate reveals that the nucleophilic attack in Step 3 would not be efficient in the reaction solution via desorption from the catalyst surface. This could be also confirmed from the undetected intermediates during the reaction. Therefore, the effectiveness of Step 3 will strongly depend on the proximity of two Pt atoms occupied by respective primary amine and primary imine. Unfortunately, the long average inter-distance between two surface Pt atoms (Fig. 2b), makes the nucleophilic attack in Step 3 difficult, impeding the condensation to form secondary imines on the atomic Pt sites.

While the Pt$_1$/CoBO$_x$ catalysts did efficiently catalyze the selective nitrile-to-secondary imine hydrogenation (Fig. 3). Thus, alternative active sites in Pt$_1$/CoBO$_x$ catalyst should provide the energetically favorable pathways to promote this reaction. Considering the conventional Lewis acid-catalyzed process, the abundant Lewis acidic B sites on CoBO$_x$ (Supplementary Fig. 10a, b) make us conjecture their highly possible acid-base interaction with reactants via Lewis basic N and catalyze the Step 3[19,37,38]. DFT calculations verified such a strong acid (B sites)–base (benzonitrile) interaction via a tilted configuration with an adsorption energy of −1.97 eV (Fig. 4a and Supplementary Figure. 11). The respective

adsorption energies of N-benzylideneamine and benzylamine on B sites were −2.26 eV and −2.99 eV. Due to the abundant surface B sites, N-benzylideneamine and benzylamine on two adjacent B sites with a close intimacy of ~0.45 nm have great opportunities to directly condense into secondary imines and leave one NH$_3$ on B site. Expectedly, N-benzylidenebenzylamine exhibits very weak adsorption on CoBO$_x$ (−0.79 eV, Fig. 4a) owing to the strong intermolecular tension and large steric hindrance, avoiding the over-hydrogenation of imines. NH$_3$ left on B sites with an adsorption energy of only −1.47 eV can be replaced by benzonitrile for the next cycle. Also, the neighboring N-benzylideneamine and benzylamine on adjacent B and single-dispersed Pt sites with a close distance of ~0.45 nm also can catalyze Step 3 with weaker interaction with N-benzylidenebenzylamine (−0.41 eV). However, the much stronger adsorption of NH$_3$ on Pt (−2.95 eV) indicates the difficulty in releasing the active sites for the next cycle, compared to that on B sites, suggesting the energetically favorable pathways on the adjacent Lewis acidic B sites.

To experimentally confirm their adsorption behaviors, the Pt$_1$/CoBO$_x$ catalysts were treated by benzonitrile, benylamine and N-benzylidenebenzylamines at 90 °C for 1 h, respectively. Their interactions were monitored by the XPS N1s peak (Fig. 4b). Similar to the fresh catalyst, Pt$_1$/CoBO$_x$ treated by N-benzylidenebenzylamines showed a bare N1s signal. While the strong N1s peaks were detected for catalysts treated by benzonitrile and benzylamine. Thus, revealed from the XPS and DFT data, the Pt$_1$/CoBO$_x$ catalyst showed a weak interaction with secondary imines but strong adsorption of nitriles and primary amines.

Meanwhile, the CoBO$_x$ nanosheets (Supplementary Fig. 12) were prepared in the absence of the Pt precursor and their catalytic performance was evaluated to further identify the function of single-atom Pt. As shown in Supplementary Fig. 13, the CoBO$_x$ nanosheets exhibited no catalytic activity for the hydrogenation of benzonitrile even after 20 h under the same reaction conditions. Therefore, the atomically dispersed Pt sites are recognized to activate H$_2$. Despite Pt sites might be occupied by reactants (nitriles) or intermediates (primary imines/amines, and NH$_3$), the single-dispersed Pt bonded with Co atom can allow the

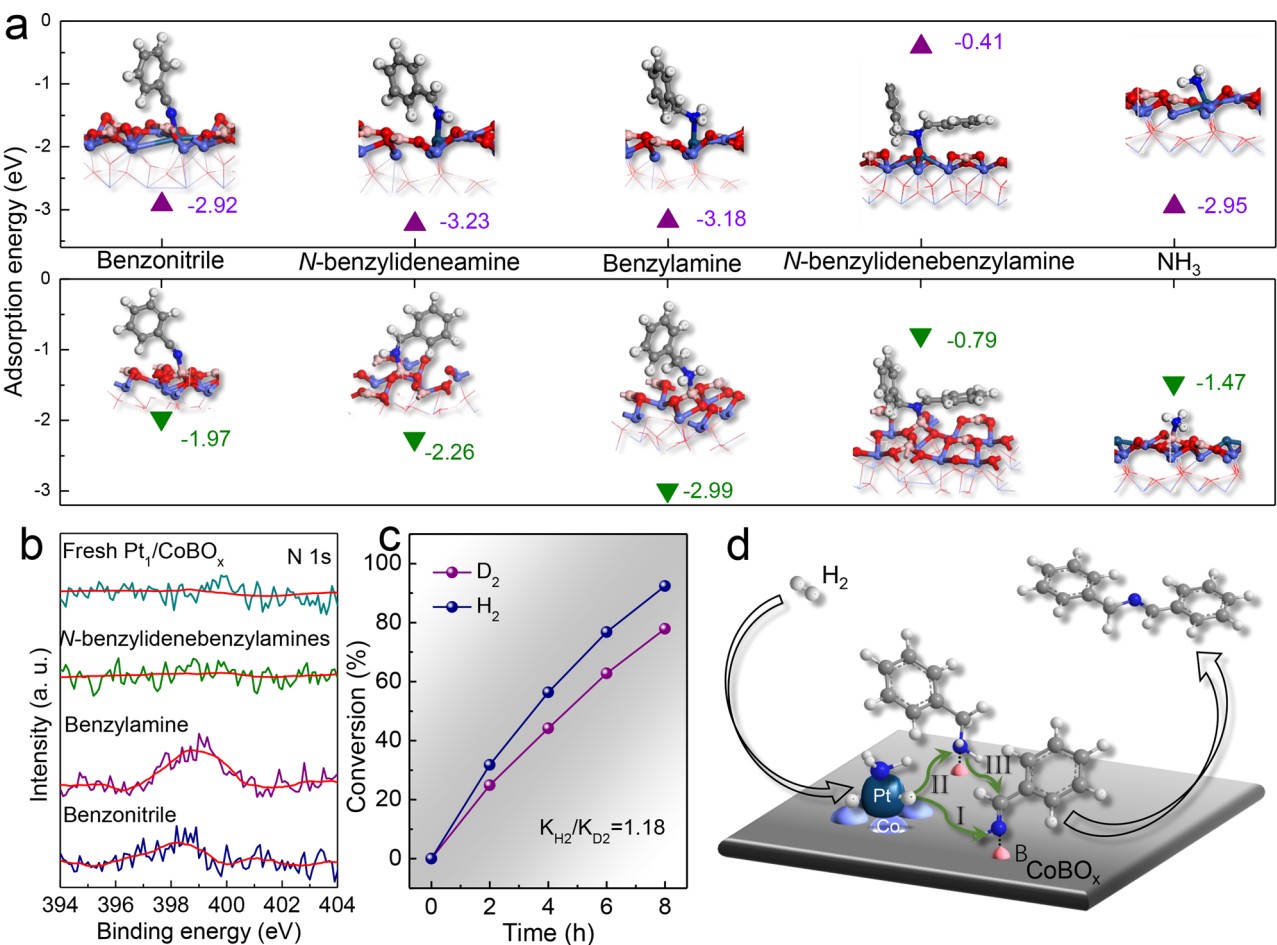

**Fig. 4 Catalytic mechanism analysis. a** Adsorption of benzonitrile, dibenzylamine, benzylamine, *N*-benzylidenebenzylamine, and $NH_3$ on catalysts. Cyan: Pt, Light blue: Co, Red: O, Pink: B, Dark blue: N, Dark grey: C, Light grey: H. **b** XPS analysis of catalysts before and after treatments with various molecules. **c** Primary isotope effects on the catalytic performance of $Pt_1/CoBO_x$. **d** Proposed catalytic mechanism. Cyan: Pt, Light blue: Co, Red: O, Pink: B, Dark blue: N, Dark grey: C, Light grey: H.

accessibility of $H_2$ due to its small size. Different from the heterolytic dissociation on single-dispersed metal catalyst[26], herein, hydrogen would undergo homolytic dissociation on the $Pt_1/CoBO_x$ catalyst into active hydrogen species due to the near metallic state of Pt as well as the presence of vicinal two metal atoms Pt and Co. The kinetic isotope effect with the use of $D_2$ in the benzonitrile hydrogenation was used to examine the proposed hydrogen homolytic dissociation pathway. The hydrogenation was slightly slowed down by a factor of 1.18 as a result of the zero-point energy difference between isotopic isomers, similar to the conventional metal catalysts (Fig. 4c), indicating the homolytic dissociation of hydrogen on the surface of $Pt_1/CoBO_x$ catalysts[26]. Then, the activated hydrogen species are spilled from the single-dispersed Pt to B sites for hydrogenation. This spillover was confirmed by the color change of the mixture of $WO_3$ and $Pt_1/CoBO_x$ under 1 MPa $H_2$ at 30 °C for 0.5 h (Supplementary Fig. 14).

Therefore, both experimental and theoretical results stated the importance of the spatial intimacy of single-dispersed Pt sites and interfacial Lewis acidic B sites for such a highly selective nitrile-to-secondary imine hydrogenation. Strong adsorption of primary amines on both single-atom Pt and Lewis acidic B sites can effectively avoid the release of by-product amines. While weak adsorption of secondary imines on $Pt_1/CoBO_x$ can suppress their over-hydrogenation. As illustrated in Fig. 3d, our approach includes several processes as the elementary steps of the catalytic cycle: the homolytic hydrogen activation on single-dispersed Pt

site in the presence of Pt–Co bond (I); the spillover of the active hydrogen species from Pt to Lewis acidic B sites for hydrogenating benzonitrile into *N*-benzylideneamine and benzylamine (II); their condensation into secondary imines on adjacent Lewis acidic B sites (III); and easy desorption of secondary imines from the catalyst surface due to the strong intermolecular tension and steric hindrance of imines (IV). The much stronger adsorption of nitriles than that of $NH_3$ on the B site guarantees the recovery of active sites for the next cycle. The $Pt_1/CoBO_x$ catalyst also showed good performance for several substituted nitriles (Table 2).

## Discussion

We have demonstrated the importance of the spatial intimacy of the binary active sites in a well-designed atom-dispersed metal catalyst for the complex catalytic reactions, which may be very different from the general metal SAC catalysts as well as the metal nanoparticle catalysts. The $Pt_1/CoBO_x$ catalysts with the spatially well-organized binary active sites of the single-dispersed Pt sites and interfacial Lewis acidic B sites have been rationally designed and dedicatedly synthesized to realize the highly selective nitrile-to-secondary imine hydrogenation. The long inter-distance between two single-dispersed Pt sites can enable the hydrogenation but inhibit the consequent condensation due to the spatial limitation. While, the adjacent interfacial Lewis acidic B sites and the neighboring single-dispersed Pt and interfacial Lewis acidic B sites with

**Table 2 Hydrogenation of various substituted nitriles[a].**

| Entry | Substrates | Products | Conv. (%) | Sel. (%) |
|---|---|---|---|---|
| 1 | | | 100 | 98.6 |
| 2 | | | 100 | 97.3 |
| 3 | | | 100 | 99.1 |
| 4 | | | 100 | 98.3 |
| 5[b] | | | 86 | 90.6 |
| 6[b] | | | 73 | 93.9 |

[a]Reaction conditions: substrates (0.5 mmol), isopropanol (2 mmol), catalysts (5 mg), 90 °C, 1 MPa $H_2$ and 6 h.
[b]100 °C and 18 h.

the intimate distances of ~0.45 nm can guarantee both the hydrogenation and condensation to successfully achieve this selective transformation. Experimental results and DFT calculations suggest the sequence of the hydrogen activation at single-dispersed Pt sites, hydrogen spillover from Pt to $CoBO_x$ supports, nitrile hydrogenation and condensation at the adjacent Lewis acidic B sites on $CoBO_x$ provides the energetically favorable pathways for this highly selective nitrile-to-imine transformation. This strategy provides insights into this previously difficult transformation by the rationally designed heterogeneous catalysts.

## Methods

**Preparation of $Pt_1/CoBO_x$ catalysts**. The $Pt_1/CoBO_x$ catalysts were prepared by a wet chemical process. Initially, 3 mmol of $Co(NO_3)_2 \cdot 6H_2O$ and 0.015 mmol of $(NH_3)_4Pt(NO_3)_2$ were completely dissolved in 280 mL $H_2O$. Then, 20 mL of freshly prepared $NaBH_4$ solution (0.375 M) was quickly added under vigorous stirring at room temperature for 1 h. After 2 h aging, the $Pt_1/CoBO_x$ catalysts were alternatively washed by water and ethanol for three times. Finally, the $Pt_1/CoBO_x$ catalysts were dried at 60 °C for 6 h.

**Preparation of Pt/C and PtCo/C catalysts**. The Pt/C catalysts were prepared by the chemical coprecipitation method. Firstly, 200 mg of carbon black were washed by hydrochloric acid (1 M) and acetone, and then dispersed in 20 mL of $H_2O$. The dispersion solution was stirred for 1 h at room temperature after adding 5 mL of $(NH_3)_4Pt(NO_3)_2$ solution (Pt: 0.32 mg mL$^{-1}$). Then, 15 mL of aqueous urea solution (10 mg mL$^{-1}$) was added into the above solution. Afterward, the reaction temperature was increased to 70 °C for 2 h. Next, 10 mL of the ice-cold fresh $NaBH_4$ solution (1 mg mL$^{-1}$) was added for another 0.5 h reaction at room temperature. Finally, the Pt/C catalysts were collected after thorough washing and drying at 60 °C.

The PtCo/C catalyst was prepared by the same process only by changing the 5 mL of $(NH_3)_4Pt(NO_3)_2$ solution to 5 mL of $(NH_3)_4Pt(NO_3)_2$ and $Co(NO_3)_2$ mixture solution.

**Characterizations**. TEM studies were conducted with a Hitachi HT-7700 transmission electron microscope with an accelerating voltage of 120 kV. XPS spectra was acquired using a Thermo Electron model K-Alpha with Al $K_\alpha$ as the excitation source. High-resolution TEM was conducted on a Titan Cubed Themis G2 300 (FEI) aberration-corrected scanning transmission electron microscope.

**Catalytic hydrogenation**. The nitrile hydrogenation was carried out in a stainless-steel autoclave equipped with the pressure control system. For a typical catalytic reaction, 1 mmol of benzonitrile and 5 mg of catalysts were mixed in 2 mL of isopropanol. The reactions were performed after charging with 1 MPa $H_2$ at 90 °C. After the reaction, the products were analyzed by GC–MS and GC.

**Adsorption experiments**. The adsorption experiments were operated in glass bottles (10 mL). Initially, 5 mg of $Pt_1/CoBO_x$ and 1 mmol of adsorbates (benzonitrile, N-benzylidenebenzylamine, or benzylamine) were mixed in 2 mL of isopropanol. The mixtures were stirred at 90 °C for 2 h. Afterwards, the treated $Pt_1/CoBO_x$ catalysts were collected by centrifugalization, washed by ethanol for three times to remove the un-adsorbed molecules, and dried by vacuum for 10 h. Finally, the treated $Pt_1/CoBO_x$ catalysts were examined by XPS.

**Cyclic experiments**. The cyclic experiments were also carried out in a stainless-steel autoclave equipped with the pressure control system. For a typical catalytic reaction, 1 mmol of benzonitrile and 5 mg of catalysts were mixed in 2 mL of isopropanol. The reactions were performed after charging with 1 MPa $H_2$ at 90 °C after 8 h. After the reaction, the $Pt_1/CoBO_x$ catalysts were collected by centrifugalization and used for the next cycle without any treatments. The products were analyzed by GC.

## Data availability

The authors declare that the main data supporting the findings of this are available within the article and Supplementary information from the corresponding author upon reasonable request. Source data are provided with this paper.

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

## Acknowledgements

We acknowledge the National Natural Science Foundation of China (21872109 and 22002115). S. Zhang is supported by the Youth Talent Support Project from China Association of Science and Technology, and the Natural Science Basic Research Plan in Shaanxi Province of China (2019JQ-039). The calculations were performed by using the HPC Platform at Xi'an Jiaotong University and National Supercomputing Center in Tianjin. We gratefully acknowledge XAS measurements at the BL14W1 beamline of the Shanghai Synchrotron Radiation Facility.

## Author contributions

S.Z., Z.X., and Y.Q. designed the studies and wrote the paper. S.Z., Y.Z., and M.Z. performed most of the experiments. Z.X. carried out the DFT calculations. S.Z., Z.X., and Y.Q. performed the data analysis. All authors discussed the results and commented on the paper.

## Competing interests

The authors declare no competing interests.
