## [Peer Review File · Nature Communications]

Reviewers' comments:

Reviewer #1 (Remarks to the Author):

This paper by Prof. Qu and co-workers reports a very interesting study of a Pt/CoBOx catalyst to enable hydrogenation of benzonitriles to secondary amines. The authors have synthesized and characterized the catalysts (Pt/CoBOx, Pt/C), tested the catalytic performance of these catalysts by hydrogenating benzonitrile (BN) and related chemical species, and performed theoretical calculations by way of Density Functional Theory (DFT).

The work claims that the single-dispersed Pt sites and interfacial Lewis acidic B sites together are the cause of selective hydrogenation to secondary amines, because of activation of the very selective nitrile to imine transformation on CoBOx sites.

The authors have no discussion or introduction on the well-established fact that Co metal catalysts are excellent catalysts to hydrogenate selectively to primary amines which may be shifting the product distributions.

The Pt1-Co4 sites of Fig 1f do not match the atomistic diagram of Scheme 1, as Scheme 1 is misleading and doesn't show the very important Pt-Co bonds.

There are also a number of typos such as 'hemolytic' and 'spatial' that should have been proofread before submitting to peer review.

The authors are encouraged to restructure their manuscript with a revised discussion, introduction, and abstract but as written the data doesn't support the hypothesis stated including the title, as the critical Pt1-Co4 sites (likely driving the product distribution shift) are not emphasized, and instead the authors indicate the B sites as important in their rational catalyst design.

It is for these reasons that I cannot support the publication of this work in its current state.

Reviewer #2 (Remarks to the Author):

This manuscript reports a strategy of catalysis by design using single site concepts.

Even if I consider this manuscript original there are too many problems to make it attractive enough for a catalysis community.

- the first problem is the complexity of the strategy, which requires cascade reactions involving hydrogenation followed by condensation starting from nitrile to finish to secondary imine.
- the complexity of the catalyst Pt/CoBO4 which is supposed to be very well defined (including STEM, EXAFS, DFT..)

- the relatively bad presentation of the results (see for example figure 1 and figure 2.

All in all this paper could be a very interesting contribution to cascades reactions using single site concepts but the authors have not simplified the strategy, the preparation and the presentation. at this stage i am enclined to refuse such paper asit stands.

it could be improved quite a lot if these various points are clearly modified.

Reviewer #3 (Remarks to the Author):

The authors described a bifunctional catalyst for selective formation of imines from nitriles. The characterization is well done and the hypothesis to determine the effectiveness of binary system is adequate.

1.- Regarding the catalytic activity I miss a comparative study with other systems to evaluate the catalytic performance of Pt/Box.

2.- The hydrogenation of substrates with halide groups gives rise to dehalogenated products?.

3.- Details of characterization data for the resulting imines should be provided.

4.- Recyclability experiments should be explain in detail and hot filtration experiments are necessary to evaluate the recyclability of catalytic system and discard any leaching for the active species.

Reviewer 1

This paper by Prof. Qu and co-workers reports a very interesting study of a Pt/CoBO_x catalyst to enable hydrogenation of benzonitriles to secondary amines. The authors have synthesized and characterized the catalysts (Pt/CoBO_x, Pt/C), tested the catalytic performance of these catalysts by hydrogenating benzonitrile (BN) and related chemical species, and performed theoretical calculations by way of Density Functional Theory (DFT). The work claims that the single-dispersed Pt sites and interfacial Lewis acidic B sites together are the cause of selective hydrogenation to secondary amines, because of activation of the very selective nitrile to imine transformation on CoBO_x sites.

We thank very much for the suggestions raised by the reviewer. We would like respond the comments as bellow.

1. The authors have no discussion or introduction on the well-established fact that Co metal catalysts are excellent catalysts to hydrogenate selectively to primary amines which may be shifting the product distributions.

Response: We thank you the comments of reviewer.

We would like to emphasize that the purpose of this research is to break the limitation of metal single-atom catalysts for the complicated cascaded reactions, which generally involves the multiple reactants and steps. As a typical cascaded reaction, the selective transformation of nitriles to secondary imines, as a green process to replace the consumption of relatively expensive amine, herein, was selected as a model reaction. Thus, our focus wasn't placed on the selective hydrogenation of nitriles.

The challenge of the model reaction to obtain the secondary imines catalyzed by single-atom catalyst is that the complex reaction process through the nucleophilic attack on the electron-deficient carbon of the *in-situ* generated primary imines by primary amines from nitrile hydrogenation (Scheme 1c). However, the highly isolated single metal sites occupied by one adsorbed molecule might not be efficient or effective for those reactions. Therefore, the part of Introduction is to provide a feasible

approach to solve this challenge.

We do agree that Co metal catalysts are excellent catalysts to selectively hydrogenate nitriles to primary amines, which may be shifting the product distributions. This is out of scope of our manuscript.

2. The Pt₁-Co₄ sites of Fig 1f do not match the atomistic diagram of Scheme 1, as Scheme 1 is misleading and doesn't show the very important Pt-Co bonds.

Response: Thank you for your reminding. We have changed the atomistic diagram of Scheme 1d to match the structure of Pt₁-Co₄ sites of Figure 1f.

Scheme 1. (d) Spatial intimacy of binary active-sites of single-atom Pt sites and interfacial Lewis acidic B.

The key point to achieve the selective hydrogenation of benzonitriles to secondary imines is the adjacent B...B on CoBO_x and neighbouring Pt...B pairs with close intimacy of ~0.45 nm. While, the present of Pt-Co bonds in Pt₁/CoBO_x is beneficial for the dissociation of H₂ to generate the activated hydrogen, which has been demonstrated and described in the manuscripts. The Scheme 1 is to express the spatial intimacy of binary active-sites, helping the reader to understand and imagine it more easily.

3. There are also a number of typos such as 'hemolytic' and 'spatial' that should have been proofread before submitting to peer review.

Response: We have double checked our manuscripts and modified the typos.

4. The authors are encouraged to restructure their manuscript with a revised discussion, introduction, and abstract but as written the data doesn't support the hypothesis stated

including the title, as the critical Pt₁-Co₄ sites (likely driving the product distribution shift) are not emphasized, and instead the authors indicate the B sites as important in their rational catalyst design.

Response: We thank the reviewer's suggestions in our writing of the manuscript. We have improved our writing to support our proposed catalytic mechanism. However, the Pt₁-Co₄ sites play the critical function for hydrogen activation in the selective hydrogenation of benzonitriles to secondary imines, instead of the driving the product distribution shift. The reasons are as follows: (1) the huge steric hindrance on Pt₁-Co₄ sites makes it difficult to bind with two reactant molecules simultaneously; (2) the Co atom in Pt₁-Co₄ sites is located in the second atom layer or subsurface, rather catalyst surface (Figure 1f). Also, Co atom is saturated by O and Pt atoms, leading to the blocked sites to bind with other reactant molecules.

Figure 1f. Calculated model of Pt₁/CoBO_x. The cyan, blue, red and pink balls represent the Pt, Co, O and B atoms, respective.

Objectively, it's difficult to precisely synthesize the Pt₁-Co₄ clusters on other supports and experimentally demonstrate their influence on the catalytic performance. Herein, the PtCo/C catalysts with 1:4 atomic ratio of Pt:Co and 1 wt.% Pt-loading (Figure S7) were chosen as the nearest approximation model catalyst to understand the roles of Co incorporated with Pt on the catalytic behavior. As shown in Figure 2, the

PtCo/C catalyst exhibited the improved catalytic activity compared with the Pt/C catalyst, which was still much slower than that of the Pt₁/CoBO_x catalysts. Most importantly, the selectivity of PtCo/C towards secondary imines *N*-benzylidenebenzylamine was similar to Pt/C and greatly lower than that catalyzed by Pt₁/CoBO_x at the end of the reaction under the same conditions. While the selectivity of benzylamines was increased to 30% for the PtCo/C catalysts. Obviously, integration of Co and Pt did not shift the selectivity of secondary imines herein. Therefore, these experimental evidences proved that the Pt₁-Co₄ sites in Pt₁/CoBO_x was not the main factor for the selective transformation of benzonitriles into secondary imines.

Figure 2 | Catalytic performance of Pt₁/CoBO_x, Pt/C and PtCo/C. (a) Time course of benzonitrile conversions. (b) Benzonitrile conversion vs. selectivity of secondary imines. (c) Final yield of various products. (d) Stability of Pt₁/CoBO_x. **Reaction conditions:** benzonitrile (1 mmol), isopropanol (2 mL), catalysts (5 mg), 90 °C and 1 MPa H₂. The reaction time of stability test was 8 h.

It is for these reasons that I cannot support the publication of this work in its current state.

Response: Thank you for your comments. We are receptive to the constructive comments and valuable suggestions. More controllable experiments have been added in our revised manuscript. The description of this manuscript has also been modified to illustrate our views more logically. With the positive and confirming results from these additional results, we have become more assertive of our work, both its significance, novelty, and conclusions.

Reviewer 2

This manuscript reports a strategy of catalysis by design using single site concepts. Even if I consider this manuscript original there are too many problems to make it attractive enough for a catalysis community.

We thank the reviewer's for raising the comments on the manuscript. We would like address his/her doubts on our studies.

1. The first problem is the complexity of the strategy, which requires cascade reactions involving hydrogenation followed by condensation starting from nitrile to finish to secondary imine.

Response: Generally, imines are predominantly produced from the condensation of primary amines and aldehydes or ketones in the presence of an acid catalyst through thermodynamic control. Meanwhile, many catalytic and non-catalytic processes, such as the oxidative dehydrogenation of secondary amines using O₂, self-condensation of amines and condensation from alcohols and amines have also been widely explored to prepare imines in the more effective manner. However, all these above-mentioned processes are inevitably associated with environmentally unfriendly reaction conditions and poor selectivity towards imines, while the relatively expensive amines are massively utilized as essential raw materials for the formation of imines. Therefore, the selectively sequential hydrogenation-condensation of nitrile into secondary imines is highly desirable since it represents a facile, green, and valuable process.

First of all, the transformation from nitrile to secondary imines is a one-pot process with multiple cascaded steps. Therefore, **this strategy is not technically complicated** compared with the other strategy to obtain secondary imines.

Secondly, our design is **not complex**. In short, **we construct the dual active sites in catalysts and make them as close as possible**, thereby leading to the selective hydrogenations, as shown in Scheme 1. It's not a complex strategy.

Besides, **the preparation of Pt₁/CoBO_x catalysts is also not complexity via a facile one-step wet chemical process at room temperature.**

Overall, supported single-atom catalysts exhibited the highest Pt atom utilization rate. The binary active-sites for sequential hydrogenation-condensation of nitriles into secondary imines is high-efficiently achieved by the spatial intimacy of single atom Pt and Lewis acid B sites. Actually, it is a meaningful strategy to obtain an efficiency catalyst for secondary imines synthesis from low-cost nitriles via a green process.

2. The complexity of the catalyst $\text{Pt}_1/\text{CoBO}_4$ which is supposed to be very well defined (including STEM, EXAFS, DFT.)

Response: Initially, the $\text{Pt}_1/\text{CoBO}_x$ catalyst were prepared by a facile wet chemical process at room temperature without complex and/or specific operation process. Thus, this is a very simple process to obtain the $\text{Pt}_1/\text{CoBO}_x$ catalysts. However, to deeply understand the unique catalytic performance, the fine structure of $\text{Pt}_1/\text{CoBO}_x$ catalyst should be clearly defined by various characterization techniques. As mentioned by reviewer, we did have STEM, EXAFS and DFT in our submissions to characterize the manuscript. Those characterizations on the catalyst structures help us the further and deep understanding the catalytic mechanism as well as provide the feasible approach for the catalyst design for other catalytic reactions.

3. The relatively bad presentation of the results (see for example figure 1 and figure 2.

Response: We are sorry for the unidentified comment. The bad presentation means the poor quality of our pictures or what else? We have modified the presentation of Figure 1 and Figure 2, as follows:

Figure 1 | Characterizations of the Pt₁/CoBO_x catalysts. (a) Dark field TEM and (b) HAADF-STEM images of Pt₁/CoBO_x. (c) XANES and (d) EXAFS of Pt₁/CoBO_x, Pt foil and PtO₂ powder. (*k*³ weighted data) (e) Curve fittings of EXAFS data. (f) Calculated model of Pt₁/CoBO_x. The cyan, blue, red and pink balls represent the Pt, Co, O and B atoms, respective.

Figure 2 | Catalytic performance of Pt₁/CoBO_x, Pt/C and PtCo/C. (a) Time course of benzonitrile conversions. (b) Benzonitrile conversion vs. selectivity of secondary imines. (c) Final yield of various products. (d) Stability of Pt₁/CoBO_x. **Reaction conditions:** benzonitrile (1 mmol), isopropanol (2 mL), catalysts (5 mg), 90 °C and 1 MPa H₂. The reaction time of stability test was 8 h.

4. All in all this paper could be a very interesting contribution to cascades reactions using single site concepts but the authors have not simplified the strategy, the preparation and the presentation.

Response: We thank you for using “a very interesting contribution...” on our study. However, we do not agree with the complicated strategy. We have given the detailed response in the Comment 1 to explain our design strategy as well as our motivations. We believe it’s unfair to claim the complex or simplified strategy. The useful strategy is the standard for a scientific issue. What we have done in this work is to design the catalysts to solve the challenges for the state-of-the-art research.

If carefully examine our design, it might not be proper to call it as “complex” strategy. As shown in Scheme 1d, the dual active sites with the spatial intimacy in the metal SAC is designed to realize the selective cascade reactions, which is difficult for metal SAC catalysts and metal nanoparticles supported catalysts. There are only two points: (1) dual active site and (2) proper spatial distance of dual active sites. Moreover, the synthesis of the catalysts is very simple and facile through a one-step wet chemical process by $\text{Co}(\text{NO}_3)_2$, $(\text{NH}_3)_4\text{Pt}(\text{NO}_3)_2$ and NaBH_4 without any surfactants or additives. Thus, it’s easy to understand our concept of catalyst design for the catalysis community. At this stage I am inclined to refuse such paper as it stands.

It could be improved quite a lot if these various points are clearly modified.

Response: Thank you for your commons. We are receptive to the constructive comments and valuable suggestions. The description of this manuscripts has also been modified to illustrate our views more logically. More controllable experiments have been added in our revised manuscript. We have become more assertive of our work, both its significance, novelty, and conclusions.

Reviewer 3

The authors described a bifunctional catalyst for selective formation of imines from nitriles. The characterization is well done and the hypothesis to determine the effectiveness of binary system is adequate.

We thank the reviewer for his/her positive comments on our manuscript and we would like address those comments raised by the reviewer.

1. Regarding the catalytic activity I miss a comparative study with other systems to evaluate the catalytic performance of Pt/BO_x.

Response: As shown in Figure 2, the Pt₁/CoBO_x catalysts exhibited the simultaneously enhanced activity and selectivity, compared with the Pt/C benchmark catalysts. Meanwhile, the catalytic performance of PtCo/C catalysts have been added in our revised manuscript to further evaluate the catalytic performance of Pt₁/CoBO_x catalysts.

Ziegler catalysts and Raney catalysts are the earliest catalysts for hydrogenation of nitriles; however, their catalytic activity and stability are relatively poor. Recently, many efforts have been focused on the homogeneous catalysis based on 3d transition metal owing to their controlled selectivity to primary amines, secondary amines and secondary imines. In heterogeneous catalysis, few cases can achieve the hydrogenation nitriles. We summarized the catalytic performance of hydrogenation nitriles based on various heterogeneous catalysts. As shown in Table S1, various catalytic systems can catalyze the nitriles to primary amines or secondary amines. However, few cases can achieve the selective hydrogenation nitriles to secondary imines. Compared with the relatively successful Pt/Ni-MOF catalyst (Supplementary Table 2, Entry 1 and 12), the Pt₁/CoBO_x exhibited obviously enhanced catalytic activity with no reduced selectivity. Therefore, combining with control experiments and previous reports, the Pt₁/CoBO_x catalyst undoubtedly exhibited improved catalytic activity for hydrogenation of nitriles and selectivity of secondary imines.

Figure 2 | Catalytic performance of Pt₁/CoBO_x, Pt/C and PtCo/C. (a) Time course of benzonitrile conversions. (b) Benzonitrile conversion vs. selectivity of secondary imines. (c) Final yield of various products. (d) Stability of Pt₁/CoBO_x. **Reaction conditions:** benzonitrile (1 mmol), isopropanol (2 mL), catalysts (5 mg), 90 °C and 1 MPa H₂. The reaction time of stability test was 8 h.

2. The hydrogenation of substrates with halide groups gives rise to dehalogenated products?

Response: Thank you for this remind. In our experiments, the hydrogenation of substrates with halide groups gives rise to dehalogenated products. And their main by-products of hydrogenation of 4-chlorobenzonitrile/4-bromobenzonitrile are the benzylamine and 4-chlorobenzylamine/4-bromobenzylamine.

3. Details of characterization data for the resulting imines should be provided.

Response: Thank you for this constructive suggestion. The relevant mass spectrum data of the secondary imines have been added in supporting information, as follows:

Entry 1

Entry 2

Entry 3

Entry 4

Entry 5

Entry 6

4. Recyclability experiments should be explain in detail and hot filtration experiments are necessary to evaluate the recyclability of catalytic system and discard any leaching

for the active species.

Response: The detail recyclability experiments have been added in our revised manuscripts. In our experiments, the Pt₁/CoBO_x catalysts can be easily recycled by centrifugalizing. Therefore, the hot filtration process is not involved in our experiment. Meanwhile, the total leaching Pt active species was 0.5 wt.% for the 4 cycles of reaction by ICP-OES analysis. Thus, the good stability of Pt₁/CoBO_x catalyst could be also confirmed from the Pt leaching results. The related description was shown as follows:

In addition, the Pt₁/CoBO_x catalysts also exhibited the preserved performance at least for four consecutive cycles (Figure 2d). After the reaction, the Pt₁/CoBO_x catalyst could be easily recycled by centrifugal separation and reused for the next cycle without any treatment. And the concentration of Pt in the reaction solution was 30 ppm by inductively coupled plasma optical emission spectrometer analysis. Thus, the total leaching Pt active species was 0.5 wt.% for the 4 cycles of reaction, which revealed the good catalytic stability. Meanwhile, the spent catalysts reserved the initial morphological features, further indicating their structural robustness (Figure S6).

REVIEWER COMMENTS

Reviewer #3 (Remarks to the Author):

The authors have included in the manuscript the modifications that I had suggested in my first review and I am satisfied with the changes made.

Reviewer #4 (Remarks to the Author):

This manuscript by Prof. Qu et al has investigated the Pt₁/CoBOx catalyst for the hydrogenation and condensation of nitriles to secondary imines. The topic of the work is interesting; however, the manuscript needs to be revised before publication can be recommended in such a high level of journal as Nature Communications. Following major points need to be addressed:

1. Being lack of any reduction process in the synthesis, the change of Pt from 2+ in the precursor to near metallic oxidation state in the catalyst needs to be explained.
2. As cobalt can catalyze the hydrogenation step of nitrile, the blank experiment should also be performed using CoBOx only. Considering the initial loading of Pt is 0.75 wt.%, and a leaching of 0.5 wt.% occurs after 4 cycles of reaction, but the conversion remains up to 86%.
3. There are details missing in the experimental section, for example, the adsorption experiments, the cyclic experiments, etc. The lack of information creates difficulty to review the results.
4. Since the paths for Pt-Pt, and Pt-Co are difficult to be clearly resolved, the Feff paths used for fitting (Pt-Co, Pt-Pt path, and Pt-O) should be provided. Also, the standard deviation, the reduced chi-square and the k₃-weighted k plot should be given to evaluate the quality of fitting.
5. The authors introduced PtCo/C as a reference catalyst, however, the characterization of this catalyst is poor. Is the Pt supposed in a Pt₁-Co₄ structure?

Minor: there are still typos and mistakes in the manuscript: "which revealed the weak meal loss"
"2.6×10¹² photons per second."

Reviewer 3

This manuscript by Prof. Qu et al has investigated the Pt₁/CoBO_x catalyst for the hydrogenation and condensation of nitriles to secondary imines. The topic of the work is interesting; however, the manuscript needs to be revised before publication can be recommended in such a high level of journal as Nature Communications. Following major points need to be addressed:

1. Being lack of any reduction process in the synthesis, the change of Pt from Pt²⁺ in the precursor to near metallic oxidation state in the catalyst needs to be explained.

Response: Thank you for your reminding. The Pt₁/CoBO_x catalysts were prepared by a wet chemical process in the NaBH₄ solution. Therefore, the NaBH₄ solution provides a reductive environment during synthesis. The standard reduction potential of Pt²⁺ to Pt⁰ is 1.18 eV, which is higher than the value of -0.28 eV of Co²⁺ to Co⁰. Thus, the Pt²⁺ ions are more easily reduced than Co²⁺ ions under the same reduction conditions, possibly resulting in the formation of specific Pt₁Co₄ sites. Meanwhile, both of our experimental and theoretical results provided the strong evidences to confirm the near metallic state of Pt in the Pt₁/CoBO_x catalysts.

We have added the relevant description in the revised manuscript to clear this question (Page 6, Line 3-6), as following:

“This phenomenon could be attributed to the reductive environment of synthetic solution in the presence of NaBH₄ as well as the much easier reducibility of Pt²⁺ ions than that of Co²⁺ ions according to their standard reduction potentials.”

2. As cobalt can catalyze the hydrogenation step of nitrile, the blank experiment should also be performed using CoBO_x only. Considering the initial loading of Pt is 0.75 wt.%, and a leaching of 0.5 wt.% occurs after 4 cycles of reaction, but the conversion remains up to 86%.

Response: As shown in supplementary Figure S12 and S13, the CoBO_x alone exhibited no catalytic activity for the hydrogenation of benzonitrile even after 20 h under the same reaction conditions. We added relevant description in our revised manuscript (Page 12, line 19-22), as following:

“Meanwhile, the CoBO_x nanosheets (Figure S12) were prepared in the absence of the Pt precursor and their catalytic performance was evaluated to further identify the function of single atom Pt. As shown in Figure S13, the CoBO_x nanosheets exhibited no catalytic activity for the hydrogenation of benzonitrile even after 20 h under the same reaction conditions.”

In addition, the description about the leaching was unprecise, leading to the misunderstanding on the stability test. The 0.5% Pt-leaching is relative to the initial amount of Pt metal in Pt₁/CoBO_x. Therefore, the Pt₁/CoBO_x with the 99.5% preserved Pt could maintain their catalytic stability. We have changed the relevant description (Page 9, Line 3-6), as following:

“Thus, the only 0.5 wt.% of Pt (relative to the initial Pt amount in Pt₁/CoBO_x) was leached from the Pt₁/CoBO_x catalyst after four cycles of the repeatedly catalytic reactions, revealing a very low metal loss and highly structural robustness of catalysts during cycling.”

3. There are details missing in the experimental section, for example, the adsorption experiments, the cyclic experiments, etc. The lack of information creates difficulty to review the results.

Response: Thank you for your reminding. We have added the relevant description in our revised manuscript (Page 16, Line 14-22, and Page 17, Line 1-5), as following:

Adsorption experiments

The adsorption experiments were operated in glass bottles (10 mL). Initially, 5 mg of Pt₁/CoBO_x and 1 mmol of adsorbate (benzonitrile, benzylamine or N-benzylidenebenzylamine) were mixed in 2 mL of isopropanol. The mixtures were stirred at 90 °C for 2 h. Afterwards, the treated Pt₁/CoBO_x catalysts were collected by centrifugalization, washed by ethanol for three times to remove the un-adsorbed molecules, and dried by vacuum for 10 h. Finally, the treated Pt₁/CoBO_x catalysts were examined by XPS.

Cyclic experiments

The cyclic experiments were also carried out in a stainless-steel autoclave equipped with the pressure control system. For a typical catalytic reaction, 1 mmol of benzonitrile and 5 mg of catalysts were mixed in 2 mL of isopropanol. The reactions were performed after charging with 1 MPa H₂ at 90 °C after 8 h. After the reaction, the Pt₁/CoBO_x catalysts were collected by centrifugalization and used for the next cycle without any treatments. The products were analyzed by GC.

4. Since the paths for Pt-Pt, and Pt-Co are difficult to be clearly resolved, the Feff paths used for fitting (Pt-Co, Pt-Pt path, and Pt-O) should be provided. Also, the standard deviation, the reduced chi-square and the k3-weighted k plot should be given to evaluate the quality of fitting.

Response: Thank you for your kind reminding. We have added the Feff paths and standard deviation, and k3-weighted k plots were also added in our revised manuscript (Figure S3, Table 1 and Table S1). And the reduced chi-square of the fittings of Pt foil and Pt₁/CoBO_x are 400.54 and 69.98,

respectively. However, the value of the reduced chi-square is not a good measure to evaluate the quality of fitting. (Vlaica, Gilberto, and Luca Olivi. "EXAFS spectroscopy: a brief introduction." *Croatica chemica acta* 77.3 (2004): 427-433.) Therefore, we prefer not add this value to Table 1.

It is very helpful, however, the relative value of the reduced chi-square between several fittings on the same data set can be used to compare the fitting. Noticing that the reduced chi-square of fitting method 1 and method 2 are 190.48 and 104.53 (Table S1), respectively, larger than 69.98 of method 3 (Table 1). This result indicates that the fitting in Table 1 is better.

Other necessary instructions are also cleared, as shown below:

As shown in dark filed TEM and HAADF-STEM images, the Pt₁/CoBO_x catalysts exhibited the atomically isolated Pt species without any apparent metallic particles/clusters on CoBO_x nanosheets. However, different from the oxidized Pt in previously reported single-atom catalysts, X-ray photoelectron spectrum (XPS) suggested a near metallic state of Pt in Pt₁/CoBO_x catalyst. Therefore, the structures of Pt₁/CoBO_x were further confirmed by XAFS. Due to the low weight percentage of Pt element (0.79 wt.%) in the Pt₁/CoBO_x catalysts, the fluorescence signal of XAFS data was collected using a solid detector with higher LOD (Limit of Detection). However, the effective fluorescence of Pt was still weak, which could be attributed to the low content of Pt as well as the interference fluorescence signals of other elements. Therefore, the collected XAFS data was noisy, especially at high k range ($k > 10 \text{ \AA}^{-1}$). The fitting of the path with heavy elements (Pt) must contain the data of k range in about $10 \sim 12 \text{ \AA}^{-1}$. We are also aware of the possible influence of this noisy on the fitting results. However, it is impossible to remove such noise without damaging the information. Therefore, the noise in the EXAFS data lead to errors of the fitted parameters and relatively large reduced chi-square. To exclude the possibility of wrong structure caused by fitting error, DFT simulation was used to further confirm the local structure of Pt₁/CoBO_x. The simulated bond lengths of Pt-Co (2.40~2.78 Å, average value of 2.65 Å) was highly consistent with the EXAFS fittings (Pt-Co: 2.61 Å). Also, one Pt atom was bonded with four Co atoms herein (Figure 1f), leaving the top of Pt for binding with oxygen, which was consistent with the fitted CN of Pt-Co (4.7) and Pt-O (1.6, Table 1).

Therefore, the single-atom Pt bonded with 4~5 Co atoms on CoBO_x could be verified by combining experimental and DFT data.

We also added necessary illustration in the revised manuscript to remind the reader (Page 6, Line 21-22), as following:

“Meanwhile, due to the low weight percentage of Pt and the interference of other elements, DFT simulation was used to further confirm the local structure of Pt in the Pt₁/CoBO_x catalysts, which

could exclude the possibility of wrong structure caused by fitting error.”

5. The authors introduced PtCo/C as a reference catalyst, however, the characterization of this catalyst is poor. Is the Pt supposed in a Pt₁-Co₄ structure?

Response: As we illustrated in the main text, it's difficult to precisely synthesize the Pt₁-Co₄ clusters on other supports and experimentally demonstrate their influence on the catalytic performance. Therefore, the PtCo/C catalysts with 1:4 atomic ratio of Pt:Co and 1 wt.% Pt-loading (Figure S7a) were chosen as the control catalyst with the nearest approximation to understand the roles of Co incorporated with Pt on the catalytic behavior. The relevant characterization of the PtCo/C catalyst was shown in Figure S7. The measured lattice spacing of 0.215 nm from the HRTEM of PtCo/C was obvious smaller than the standard Pt(111) crystalline plane (0.223 nm), but larger than the standard Co(111) plane (0.205 nm), which indicated the successful formation of PtCo alloy nanoparticles on the surface of carbon. Therefore, it is reasonable to presume that the Pt₁-Co₄ structure should be existed on the surface of PtCo/C alloy with 1:4 atom ratio of Pt:Co. However, we can not determine exactly the Pt₁-Co₄ structure in the PtCo/C catalysts.

Figure S7. Characterizations of the PtCo/C catalyst. (a) TEM and (b) HRTEM images of the PtCo/C catalyst.

We also added the description in our revised manuscript (Page 10, Line 3-6), as following:

As shown in Figure S7b, the measured lattice spacing of 0.215 nm from the HRTEM image of PtCo/C was smaller than the Pt(111) crystalline plane (0.227 nm), but larger than the Co(111) plane (0.205 nm), which indicated the successful formation of the alloyed PtCo nanoparticles on the surface of carbon.

Minor: there are still typos and mistakes in the manuscript: “which revealed the weak meal loss”

“ 2.6×10^{12} photons per second.”

Response: We are sorry for our typos and mistakes in the manuscript, and we have tried our best to correct them.

REVIEWERS' COMMENTS

Reviewer #3 (Remarks to the Author):

most of the observations made to The authors have included in the revised manuscript and it can be published.